# **Tracking the Mediterranean Abyss**

Simona Aracri<sup>1</sup>, Katrin Schroeder<sup>1</sup>, Jacopo Chiggiato<sup>1</sup>, Harry Bryden<sup>1,2</sup>, Elaine McDonagh<sup>2</sup>, Simon Josey<sup>2</sup>, Yann Hello<sup>3</sup>, and Mireno Borghini<sup>1</sup>

<sup>1</sup>ISMAR-Istituto di Scienze Marine, Venice and La Spezia <sup>2</sup>NOC-National Oceanography Institute, Southampton <sup>3</sup>Géoazur-Valbonne, France

Correspondence to: Aracri Simona (simona.aracri@ve.ismar.cnr.it)

Abstract. The abyssal velocity of the Northern Current, in the north-western Mediterranean has been estimated using for the first time MERMAIDs, i.e. submarine drifting instruments that record seismic waves. In this study the Northern Current shows an intense activity even in deep layers of the water column. Through pseudo-eulerian statistics different components of the observed variability are analysed and described, revealing the turbulent nature of the Liguro-Provençal basin abyssal airculation

5 circulation.

# Keywords

Western Mediterranean, Abyssal Circulation, Northern Current, lagrangian instruments, MERMAID

# **Copyright statement**

The article is distributed under the Creative Commons Attribution 3.0 License. Unless otherwise stated, associated published 10 material is distributed under the same licence

## 1 Introduction

The assessment of the abyssal contribution to the world ocean circulation represents one of the present challenges for oceanography. Despite the remarkable technological progresses of the last decades the deep ocean remains difficult to sample with adequate time and space resolution. Therefore it is still scarcely known and evaluating its contribution to the ocean dynamics

- 15 is a tough task. The oceanographic peculiarity of the Mediterranean Sea (Fig. 1) lies in its typical oceanic processes, e.g. deep water formation (DWF) in the Gulf of Lion (GoL) (Marshall and Schott, 1999), in the Adriatic and the Aegean Sea (Schroeder et al. (2012); Malanotte-Rizzoli et al. (2013)), deep convection in the Ligurian Sea (Sparnocchia et al., 1995) and Levantine Basin (Georgopoulos et al., 1989), the thermohaline circulation, and its reduced time-scales. Thanks to these characteristics it constitutes the perfect observatory to understand the behaviour also of the world ocean. This study analyses data from the deep
- 20 western Mediterranean Sea (WMED). The focus of this work is to study the deep circulation of the WMED and in particular

5

to observe the Northern Current (NC) evolution at depth by means of deep lagrangian floating devices, called MERMAIDs (Hello et al., 2011). The NC is a marked frontal structure (Astraldi et al., 1994) that involves the surface-intermediate layer and probably also the deep layer of the water column. It flows cyclonically along the southern European coast from the Ligurian Sea to the Alboran Sea. In the Ligurian Sea the NC is fed by the Eastern Corsica Current (ECC) and the Western Corsica Current (WCC) (Astraldi et al., 1994), see Fig. 1. Over the course of time dedicated experiments tried to depict the NC temporal and

- spatial variability (Taupier-Letage and Millot (1986); Conan and Millot (1995); Sammari et al. (1995); Alberola et al. (1995)),
  but only few refer to it as a characteristic involving the whole water column and therefore carrying deep waters (Millot, 1999).
  Most authors treat the NC as a surface-intermediate entity (Astraldi and Gasparini (1992); Robinson et al. (2001); Picco et al. (2010); Alvarez et al. (2013)). Others limit the NC influence to 1100 m depth (Taupier-Letage and Millot, 1986) and some
  touch on the possibility of the NC dragging along Western Mediterranean Deep Water (WMDW) (Conan and Millot, 1995).
- Lagrangian floats data are affected by uncertainties associated with positioning system, clock drift and unknown surface drift before submerging and after surfacing (Katsumata and Yoshinari, 2010). In previous experiments, e.g. SOFARGOS, deep floats such as RAFOS were localized at depth by triangulation using an array of moored acoustic sources, (Testor and Gascard (2003); Testor et al. (2005); Testor and Gascard (2006)). RAFOS accuracy for positioning was approximately 2 Km.
- 15 Using ARGO floats or, as in the present work MERMAIDs, the positioning is done via satellite communication instead of an acoustic triangulation. Katsumata and Yoshinari (2010) infer that the sampling covariance of the drift mean velocity estimate is inversely proportional to the drifting time: the longer the float drifts at depth the smaller is going to be the error in velocity. In the case of ARGOs, the floats are programmed to surface approximately every 10 days or 5 days in the Mediterranean depending on the chosen sampling schedule, which renders an accurate estimation of the error and of its position difficult
- 20 in most cases. MERMAIDs, in the present dataset, surface more often than ARGOs, i.e. on average every 4-5 days, so the uncertainty associated with their positioning remains an open question. MERMAIDs are programmed to ascend to the surface every 10 days, but since they are also designed to surface when they detect a seismic event, they can emerge more often, despite their default setting parameters. Differently from MERMAIDs, ARGOs have a prefixed parking depth, e.g. 350 m in the framework of MedArgo (Poulain et al., 2007), once every 10-5 days, they sink to a greater depth, i.e. 700 m or 2000 m and
- 25 then they surface performing a CTD (Conductivity Temperature Depth) profile during the upcast (Menna and Poulain (2010); Poulain et al. (2007)). In the Mediterranean Sea ARGO floats have mostly been used to investigate intermediate currents (Menna and Poulain (2010); Poulain et al. (2007); Bosse et al. (2015)), in fact the 350 m target depth corresponds to the core depth of Levantine Intermediade Water (LIW). LIW is an intermediate water formed in the Levantine Basin, in the Eastern Mediterranean Sea. During the SOFARGOS project, RAFOS floats drifting depths were set to 350 m, 600 m and 1200 m in
- 30 order to capture features of the behaviour of the entire water column. Testor and Gascard identified Submesoscale Coherent Vortices (SCV), and investigated their spatial distribution in the Algerian Basin (Testor et al., 2005) and in the north-western Mediterranean (Testor and Gascard, 2003), relating the occurrence of SCV with DWF (Testor and Gascard, 2006). SCV are long living anti-cyclonic vortices (in the northern hemisphere), which retain much of their core water mass and migrate far from their formation site (McWilliams, 1985). In Testor and Gascard (2003) the interaction between the SCV and the NC
- 35 is mentioned: the newly formed deep water "bled" into the NC, which therefore influenced the spreading of the WMDW.

Although the RAFOS measurements aimed to sample and describe SCV they also provided some insight into the behaviour of the NC. After the measurements analysed in Testor and Gascard's studies in 2002 no more lagrangian deep measurements were reported in the NC. To our knowledge no other attempt were made to quantify the NC vertical influence or strength at depth.