# Peer review of "Tracking the Mediterranean Abyss"

_Ocean Science, 2016_

## Referee Comment (RC1) · Anonymous Referee #1 · 21 Nov 2016

The tracks of sub-surface floats are used to estimate velocities in the deep layers of the northwestern Mediterranean Sea. The results show that the Northern Current appears strong and even rather turbulent in the deep sea. According to me, this paper is a very superficial and fundamentally flawed study of the deep currents in the NW Mediterranean based on a few floats. The instruments used and the methodology adopted are scarcely described. The results are not robust and do not increase significantly the knowledge on the deep circulation in the NW Mediterranean. As a result, I believe that the manuscript is well below the level of the Ocean Science Journal, and given the substantial work needed to improve it, I recommend to reject it. Some detailed comments are listed here below. They might be useful for the authors to carry a better and more rigorous analysis of the float data. 1) The title is ambitious and appealing. But what does it mean? We cannot track the abyss! The measurements are above 2000 m whereas the abyssal sea/ocean is generally much deeper. The title does not represent the work described in the paper. 2) Introduction: it is too long and not enough

focused on the deep currents in the NW Med. 3) Data and methods: More information on the MERMAIDS floats is needed. They are actually Apex floats equipped with seismic sensors. More details should be provided on how the parking velocities are computed. This problem has been addressed by Park et al. (2005) and Menna and Poulain (2010). The parking depth is essentially varying between 500 and 2000 m, you might want to exclude data above 1000 m to only consider the deep sea (e.g., exclude the Ibiza Channel). In addition, please address the fact that the float can touch the bottom. Please add the bathymetry in your Figure 2 (both versus time and in the maps) and discard points if the parking depth is close to the bathymetry. 4) Results: I don't believe that deep currents can reach 88 cm/s! Menna and Poulain (2010) in the same area have maximal currents near 350 m of 30-40 cm/s. Pseudo-Eulerian statistics with such small bins and so few data are rather useless.

---

## Referee Comment (RC2) · Anonymous Referee #2 · 1 Dec 2016

This is a short manuscript that describes the trajectories of 7 deep drifters (MER-MAIDS) in the Northwestern Mediterranean Sea. The use of these new devices is interesting and I have no doubts about the applications but, unfortunately, the material presented in the manuscript is insufficient (both in terms of quality and quantity) for an article to be published in Ocean Science. The title is appealing although too ambitious for the contents included in the manuscript. It is not possible to track the Mediterranean Abyss with only 7 drifters deployed in a specific location and time! This has to be addressed with a higher significant number of drifters, combined with observations from other sensors and numerical simulations. A few more specific comments: The introduction is not well written (different paragraphs are disconnected). Data and Methods: More details are needed to better understand the functioning of MERMAIDS, as this is a non standard instrument. P. 4: detection of a seismic wave: add a reference or explain. Results: 'It is assumed that the instrument descends vertically': justify. I have also serious concerns about the significance of Pseudo-Eulerian statists with only one

drifter. In my opinion it does not make any sense. I must confess I am disappointed of such a low quality manuscript written by very well known authors. A more in depth analysis with a more comprehensive dataset is needed.